# RNA-Binding Proteins as Critical Post-Transcriptional Regulators of Cardiac Regeneration

**DOI:** 10.3390/ijms241512004

**Published:** 2023-07-26

**Authors:** De-Li Shi

**Affiliations:** 1Department of Medical Research, Affiliated Hospital of Guangdong Medical University, Zhanjiang 524001, China; de-li.shi@upmc.fr; 2Laboratory of Developmental Biology (CNRS-UMR7622), Institute de Biologie Paris-Seine (IBPS), Sorbonne University, 75005 Paris, France

**Keywords:** myocardial injury, cardiac regeneration, post-transcriptional regulation, RNA-binding protein, cardiac RBPome, cardiomyocyte proliferation

## Abstract

Myocardial injury causes death to cardiomyocytes and leads to heart failure. The adult mammalian heart has very limited regenerative capacity. However, the heart from early postnatal mammals and from adult lower vertebrates can fully regenerate after apical resection or myocardial infarction. Thus, it is of particular interest to decipher the mechanism underlying cardiac regeneration that preserves heart structure and function. RNA-binding proteins, as key regulators of post-transcriptional gene expression to coordinate cell differentiation and maintain tissue homeostasis, display dynamic expression in fetal and adult hearts. Accumulating evidence has demonstrated their importance for the survival and proliferation of cardiomyocytes following neonatal and postnatal cardiac injury. Functional studies suggest that RNA-binding proteins relay damage-stimulated cell extrinsic or intrinsic signals to regulate heart regenerative capacity by reprogramming multiple molecular and cellular processes, such as global protein synthesis, metabolic changes, hypertrophic growth, and cellular plasticity. Since manipulating the activity of RNA-binding proteins can improve the formation of new cardiomyocytes and extend the window of the cardiac regenerative capacity in mammals, they are potential targets of therapeutic interventions for cardiovascular disease. This review discusses our evolving understanding of RNA-binding proteins in regulating cardiac repair and regeneration, with the aim to identify important open questions that merit further investigations.

## 1. Introduction

Cardiovascular disease is the leading cause of death worldwide [1]. Myocardial infarction (MI), also commonly known as heart attack, due to the lack of oxygen supply, causes the necrosis of cardiomyocytes (CMs) and consequently leads to scar formation, reduced cardiac contractility, and heart failure [2]. The early and fast restoration of blood flow to the ischemic myocardium can limit the infarct size but also has the adverse outcomes of inducing serious damage, causing myocardial ischemia/reperfusion (I/R) injury [3]. Cardiac damage can be caused directly by ischemia due to atherosclerotic plaque disruption that induces coronary thrombosis (type I), secondarily to ischemia because of a supply–demand imbalance in oxygen in the myocardium (type II), or as a consequence of elevated troponin concentrations without myocardial ischemia [4,5]. The adult mammalian heart has no or a very limited regenerative capacity because CMs irreversibly exit the cell cycle shortly after birth. However, the heart from early postnatal mammals and from adult lower vertebrates (zebrafish and newt) can fully regenerate after ventricular resection or MI-induced injury through the proliferation of existing CMs [6,7,8]. Signals derived from multiple other cell types, such as cardiac fibroblasts and myofibroblasts, also contribute to the cardiac repair and regeneration processes [9,10,11]. Therefore, understanding the mechanisms governing cardiac regenerative capacity after heart injury could help to define strategies and technologies for inducing the formation of new CMs and promoting cardiac regeneration in mammals [12,13,14].

RNA-binding proteins (RBPs) are critically involved in the regulation of muscle cell proliferation and differentiation during development and disease [15]. They control gene expression at the post-transcriptional level, from pre-mRNA alternative splicing to mRNA transport, localization, stability, polyadenylation, and translation [15]. Moreover, in cooperation with other proteins, aggregation-prone RBPs can prevent the translation of toxic mRNAs and promote the degradation of misfolded proteins through the ubiquitin-proteasome or autophagy pathway by sequestering them into stress granules [16]. Thus RBPs make an important contribution to protein quality control. An analysis of the cardiac RBPome indicates that a large number of RBPs not only show extensive expression differences between fetal and adult hearts, but also become reactivated or repressed following heart failure, suggesting a possible contribution of these proteins in heart disease [17,18]. Indeed, at least in different animal models, RBPs are clearly involved in regulating ischemia-dependent and ischemia-independent myocardial injury. In addition, RBPs display dynamic changes in binding activity with target mRNAs between homeostatic and pathological conditions [18]. Functional studies have demonstrated that RBPs mediate injury-stimulated signals to regulate translational reprogramming, metabolic changes, and cellular plasticity, thereby contributing to the survival, growth, and differentiation of CMs in the damaged postnatal and adult mammalian hearts [19]. Due to their essential roles in heart development and disease [20,21,22], RBPs are emerging therapeutic targets for interventions in cardiovascular dysfunction [23]. In recent years, evidence is rapidly accumulating that manipulating the activity of RBPs has the strong potential to improve new CM formation and cardiac function after myocardial injury. This review discusses our evolving understanding of the mechanisms underlying RBP-regulated molecular and cellular events during cardiac repair and regeneration. It also proposes future perspectives to promote research in this field.

## 2. RNA-Binding Proteins in Cardiac Repair and Regeneration

RBPs not only display dynamic expression in fetal and adult hearts under homeostatic conditions, but also show differential activation following heart injury. Therefore, it can be expected that they have distinct functions during heart development and cardiac regeneration. Indeed, RBPs are effectors of cardiac injury stimuli and subsequently trigger a range of molecular and cellular events that influence CM proliferation with strong impacts on heart repair and function (Figure 1). In recent years, there is accumulating evidence that RBPs function as important post-transcriptional regulators of cardiac regeneration in neonatal and adult animals. The following sections discuss the known functions of RBPs in cardiac repair and regeneration in the mammalian hearts.

### 2.1. LIN28A Regulates Metabolic Reprogramming to Promote Cardiac Regeneration

LIN28A is a highly conserved small cytoplasmic RBP with functions in promoting pluripotency and regulating the transition from self-renewal to a differentiated cell fate [24]. In mice, its expression progressively decreases during postnatal heart development but is rapidly induced after heart damage [25,26]. Several studies have demonstrated that it acts as a multifaceted regulator in stimulating CM growth and protecting the heart function after cardiac injury (Figure 2). In diabetic mice with I/R-induced cardiac injury, the overexpression of LIN28A can reduce mitochondrial cristae destruction, CM apoptosis, and the myocardial infarct size likely through regulation of the insulin-PI3K–mTOR pathway [27,28]. In MI-induced cardiac injury, LIN28A can induce autophagy and inhibit apoptosis in CMs by activating Sirt1, a NAD(+)-dependent deacetylase that mediates the cellular response to inflammatory, metabolic, and oxidative stresses [29]. Beside these protective effects, LIN28A also functions to promote CM proliferation. Mouse cardiac tissue derived stem-like cells expressing LIN28A exhibit metabolic flexibility and redox regulation with increased expression of glycolytic genes; they show enhanced survival and proliferation after transplantation to the ischemic heart tissue and thus an increased ability to repair the heart [30]. Furthermore, a recent study revealed that LIN28A is involved in the control of cell cycle activity to define CM numbers in postnatal and adult hearts subjected to MI-induced injury [26]. In the adult heart, mononuclear diploid CMs are capable of division, which may contribute to CM turnover and cardiac regeneration, whereas binucleation or polyploidization prevents CM proliferation [31,32,33]. Interestingly, the overexpression of LIN28A increases the proportion of mononuclear diploid CMs and extends their cell cycle activity in the MI-injured mouse heart, thus promoting cardiac regeneration [26]. Mechanistically, LIN28A interacts with and increases the expression of IncRNA H19, which plays a role in stem cell proliferation by regulating glycolysis [34] and alleviates cardiac hypertrophy-induced heart failure [35], to reprogram CM metabolism toward glycolysis and promote the formation of mononuclear diploid CMs [26]. These observations thus demonstrate a critical role of LIN28A in metabolic reprogramming for cardiac growth. However, there is also evidence showing that LIN28A promotes pathological cardiac hypertrophy through the same mechanism. In the adult mouse heart subjected to transverse aortic constriction, LIN28A post-transcriptionally regulates the expression of mitochondrial Pck2 (phosphoenolpyruvate carboxykinase 2), which in turn promotes cardiac glycolysis and biosynthesis to induce pathological cardiac hypertrophic growth [25]. Thus, LIN28A functions as an important regulator that promotes cardiac regeneration and pathological hypertrophy. Further investigations will be necessary to provide insights into the mechanisms underlying its function in these processes. It is possible that LIN28A regulates cellular metabolism depending on the biological contexts and on the functional interaction with its targets or partners.

The function of LIN28A in different regeneration processes is often, if not always, dependent on its interaction with *let-7* miRNA. It is well established that LIN28A and *let-7* mutually suppress the post-transcriptional expression of each other in regulating cellular metabolism, as well as cell fate and growth [36,37]. In mouse cardiac tissue-derived stem-like cells, LIN28A regulates metabolic flexibility to promote survival and proliferation by suppressing *let-7* expression and activating the PKD1 (pyruvate dehydrogenase kinase 1) signaling pathway [30]. Therefore, the beneficial effects of LIN28A on cardiac regeneration are linked to the activation of let7-repressed genes. Indeed, the inhibition of *let-7* function promotes cardiac regeneration and prevents cardiac remodeling, thereby improving cardiac function after MI in mice [38,39,40]. Nevertheless, it should be mentioned that the role of *let-7* in cardiac regeneration is more complex. Although the suppression of *let-7* activity by hypoxia-induced LIN28A expression is beneficial for CM survival because this promotes glucose uptake and glycolysis, the loss of *let-7* activity during hypoxia–reoxygenation may increase CM death due to inhibition of AKT (protein kinase B) signaling [41]. Consequently, the timing of the LIN28A/*let-7* switch needs to be tightly regulated to balance CM survival and apoptosis for cardiac regeneration.

### 2.2. RNA-Binding Proteins Associated with mRNA Modifications in Cardiac Regeneration

N6-methyladenosine (m6A) represents the most abundant modification of eukaryotic mRNAs, which has important effects on gene expression in a wide variety of physiological and pathological processes by affecting multiple steps of mRNA metabolism, such as pre-mRNA alternative splicing and mRNA subcellular localization, translation, and stability [42,43]. It is well established that m6A modification is a dynamic and reversible process regulated by methyltransferases or writers (METTL3/14/16, WTAP, RBM15/15B, ZC3H3, and KIAA1429) and demethylases or erasers (FTO and ALKBH5). Moreover, the modification is recognized by several RBPs or readers, including YTHDC1/2, YTHDF1/2/3, IGF2BP1/2/3, and hnRNP A2B1/C, which function to regulate the post-transcriptional expression of their target mRNAs [42]. Evidence is emerging that m6A modification plays an important role in cardiac homeostasis and disease, making it a potential target in the therapeutic intervention for the treatment of heart failure [44,45,46]. By mediating the function of writers or erasers, several RBP readers are either positively or negatively involved in CM proliferation.

#### 2.2.1. YTHDF1 Is a Target of ALKBH5 in Promoting Cardiac Regeneration

The demethylase ALKBH5 (AlkB homolog 5) removes m6A modification from mRNAs and functions to promote cardiac regeneration. The cardiac expression of ALKBH5 progressively decreases during postnatal development but increases after apical resection injury in neonatal mice [47]. The overexpression of ALKBH5 increases CM proliferation and promotes cardiac regeneration after MI-induced injury in postnatal and adult hearts, whereas the deletion of ALKBH5 produces the opposite effects after apical resection injury [47]. Interestingly, ALKBH5 demethylates *YTHDF1* mRNA in neonatal CMs, resulting in its increased stability and translation [47]. YTHDF1 is an m6A reader that recognizes and promotes the translation of m6A-methylated mRNAs [48]. Accordingly, its increased expression triggers CMs to re-enter the cell cycle at least by promoting the accumulation of YAP1 (Yes-associated protein 1), a transcriptional cofactor regulating cardiac regeneration by activating embryonic and proliferative gene programs in CMs [47,49]. Therefore, YTHDF1 promotes cardiac regeneration through enhanced translation of its target mRNAs, but the translation and activity of itself are also post-transcriptionally regulated by ALKBH5-mediated demethylation (Figure 3A).

#### 2.2.2. YTHDF2 Mediates the Activity of METTL3 to Prevent Excessive Cardiomyocyte Proliferation

METTL3 is a pivotal catalytic subunit in the methyltransferase complex. Its expression increases in the postnatal mouse heart and following neonatal heart injury but does not change after MI-induced injury in the adult heart [50]. The overexpression of METTL3 suppresses the proliferation of primary CMs isolated from neonatal mice and inhibits cardiac regeneration in vivo, whereas the knockdown of METTL3 increases CM proliferation and promotes cardiac regeneration after MI-induced heart injury in neonatal, postnatal, and adult mice [50,51]. In the injured heart, METTL3 has been shown to trigger the methylation of *FGF16* mRNA and induce its degradation in a YTHDF2-dependent manner [50]. Consequently, YTHDF2 regulates CM growth and cardiac function through interactions with its methylated mRNA targets (Figure 3B). YTHDF2 protein shows increased expression in human heart failure tissues, as well as in hypertrophic mouse hearts [52]. The overexpression of YTHDF2 attenuates cardiac hypertrophy by inhibiting the translation of Eef2 (eucaryotic elongation factor 2) and Myh7 (myosin heavy chain 7) mRNAs in an m6A-dependent manner [52,53]. These observations suggest that YTHDF2 can prevent excessive CM growth in pathological hypertrophy. Since YTHDF2 recognizes m6A modification and reduces mRNA stability [54], it may function downstream of METLL3 to modulate the potential of CM proliferation and cardiac regenerative capacity.

#### 2.2.3. IGF2BP3 Promotes Neonatal and Adult Cardiac Regeneration

IGF2BP3 (insulin-like growth factor 2 mRNA-binding protein 3) is highly expressed in all cell types of the neonatal mouse heart, including CMs and cardiac fibroblasts, but it is progressively downregulated during postnatal development and becomes undetectable in the adult heart [55,56]. This is associated with decreased H3K27ac signals and increased H3K27me3 signals at the *IGF2BP3* locus during postnatal development, suggesting that epigenetic modifications regulate the temporal cardiac expression of IGF2BP3 [56]. The overexpression of IGF2BP3 promotes CM proliferation and cardiac regeneration in MI-injured neonatal and adult hearts [55,56], while the knockdown of IGF2BP3 prevents CM proliferation and survival in the neonatal heart [55]. The post-transcriptional mechanism by which IGF2BP3 promotes cardiac regeneration remains largely elusive. Since IGF2 plays a role in promoting CM proliferation during heart development [57,58], there is a possibility that IGF2BP3 may enhance IGF2 signaling by stabilizing *IGF2* mRNA, but this needs to be verified through further investigation. RNA-seq analysis after the knockdown of IGF2BP3 in neonatal CMs indicates that *MMP3* (matrix metalloproteinase 3) mRNA may be a potential target of IGF2BP3 regulation (Figure 3C). As with IGF2BP3, the cardiac expression of MMP3 decreases in the postnatal heart. However, IGF2BP3 stabilizes *MMP3* mRNA in an m6A-dependent manner. Moreover, the overexpression of MMP3 enhances CM proliferation, whereas the knockdown of MMP3 produces the opposite effect [55]. MMP3 is a member of the MMP family that functions to degrade components of the extracellular matrix (ECM) for tissue remodeling [59], and it is closely associated with MI and heart failure in humans and animal models [60,61,62]. Nevertheless, MMPs may also beneficially influence cardiac repair because they can facilitate the removal of necrotic CMs after MI by degrading the ECM and recruiting inflammatory cells [63]. In addition, there is evidence that MMPs can exert protective effects against hypertrophic cardiac growth [64], and it is known that the activity of MMPs may be modulated by the presence of their endogenous regulators, such as tissue inhibitors of metalloproteinases [65]. Thus, it will be of interest to examine how MMP3 functions downstream of IGF2BP3 to modulate CM proliferation. Another question concerns the reactivation of IGF2BP3 following cardiac injury. In contrast to the unchanged transcript level [56], the expression of IGF2BP3 protein seems to be potently induced at the border zone of the MI-injured neonatal heart [55]. Thus, whether injury signals trigger the enhanced translation of *IGF2BP3* mRNA in the neonatal heart merits further investigation.

### 2.3. Translational Reprogramming Mediated by PABPC and CPEB in Cardiac Hypertrophic Growth

Cytoplasmic polyadenylation regulates poly(A) tail elongation and the translation of nuclear-exported mRNAs. PABPC (cytoplasmic poly(A)-binding protein) and CPEB (cytoplasmic polyadenylation element-binding protein) are key components of the cytoplasmic polyadenylation complex [66]. PABPC protects the poly(A) tail from degradation and stimulates translation by cooperating with mRNA 5′-cap interaction factors [67]. The adult mammalian heart displays low rates of protein synthesis, which increases substantially in response to hypertrophic signals. It has been shown that this is regulated, at least partly, by poly(A) tail length changes in *PABPC1* mRNA [68]. In mice, the expression of PABPC1 protein shows progressive downregulation specifically in the heart during postnatal development. In adult human and mouse hearts, PABPC1 protein is absent because the translation of its mRNA is suppressed due to a shortened poly(A) tail length but without degradation of the transcript [68]. Intriguingly, the poly(A) tail length and translation of *PABPC1* mRNA are partially recovered in mice under physiologic and pathologic hypertrophy triggered by endurance exercise training and thoracic aortic constriction, respectively; subsequently, PABPC1 interacts with eIF4G (eukaryotic translation initiation factor 4 G) to stimulate global protein synthesis [68]. Cardiac hypertrophy may be a physiological adaptation to maintain the function or a pathological process that can lead to heart failure [69]. Importantly, the forced expression of PABPC1 can induce the physiologic growth of adult CMs in the absence of pathological hypertrophy and cardiac dysfunction [68]. This observation suggests that the dynamic regulation of PABPC1 activity and, as a result, the rates of global protein synthesis, may contribute to the beneficial CM growth in the failing heart. It also raises the interesting question of how cardiac injury triggers the cytoplasmic polyadenylation of *PABPC1* mRNA to increase global protein synthesis.

CPEB4 can act as a translational repressor by sequestering mRNAs but becomes an activator upon phosphorylation by ERK2/Cdk1 [70]. It exhibits dynamic post-transcriptional regulatory functions during the pathological growth of CMs, although its expression shows no change between normal and injured hearts [18]. Mechanistically, unphosphorylated CPEB4 binds to and sequesters *Zeb1* mRNA to inhibit its translation under homeostasis. However, CPEB4 is released from *Zeb1* mRNA during pathological growth, leading to the expression of Zeb1 protein [18]. Since Zeb1 is a transcription factor regulating the expression of cell cycle genes and promoting CM proliferation [71], the dysregulation of its activity can lead to pathological hypertrophy and decreased cardiac function. Thus, the dynamic interaction between CPEB4 and *Zeb1* mRNA should play an important role in controlling CM growth stimulated by pathological signals. At present, it is unclear how CPEB4 activity is regulated to coordinate the translation of target mRNAs in response to pathological signals. Interestingly, CPEB1, another member of the CPEB family, shows increased phosphorylation during the quiescence exit of skeletal muscle stem cells (satellite cells). This post-translational modification allows CPEB1 to promote the synthesis of myogenic factors, such as MyoD1, thus reprogramming satellite cell activation and proliferation [72]. There is also evidence that Zeb1 protects skeletal muscle from injury-induced damage and is required for the proper progression of regeneration [73]. Thus, it is possible that the dynamic regulation of CPEB RNA-binding activity may also be involved in reprogramming the translational landscape for activating CM proliferation during cardiac regeneration.

### 2.4. MBNL1 Functions as a Regulator of Cellular Plasticity during Cardiac Regeneration

It is well established that dysfunctions of the splicing regulator MBNL1 (muscle blind-like 1) due to its disrupted subcellular localization prevent the postnatal switch of adult muscle protein isoforms and are associated with skeletal muscle diseases, such as myotonic dystrophy [15]. Cardiac fibroblasts are responsible for fibrosis by depositing ECM in the heart. Myocardial damage can transform cardiac fibroblasts into myofibroblasts, which further produce cytokines and ECM, with initial benefits for heart repair but long-term detrimental effects on cardiac function [9,10]. The expression of MBNL1 increases progressively during postnatal heart development, and it promotes cellular differentiation by regulating the fetal-to-adult transition of muscle-specific alternative splicing events [74]. It is also upregulated in mouse myofibroblasts induced by MI and in fibroblasts from failing human hearts [75]. The overexpression of MBNL1 in MI-injured hearts induces the differentiation of fibroblasts into myofibroblasts and promotes the fibrotic phase of wound healing by regulating the expression of protein networks involved in cellular differentiation and cytoskeletal/matrix assembly through alternative splicing or alternative polyadenylation [76]. Therefore, MBNL1 functions as a post-transcriptional regulator of cellular plasticity during the process of heart repair (Figure 4). Furthermore, MBNL1 blocks MI-induced fibroblast proliferation and is required for maintaining the mature myofibroblast state in part by stabilizing *Sox9* mRNA [75]. Consistently, increased MBNL1 activity prevents CM proliferation, whereas MBNL1 deficiency prolongs the period for CM proliferation, thus extending the window of cardiac repair and regeneration [77]. These observations indicate that MBNL1 regulates the fate changes of multiple cell types. They also raise the possibility that manipulation of MBNL1 activity can be used for therapeutic intervention to improve wound healing and cardiac regeneration.

### 2.5. hnRNP U Prevents Cardiomyocyte Binucleation

The expression levels of hnRNP U (heterogeneous nuclear ribonucleoprotein U) progressively decrease during postnatal development. It regulates the cardiac-specific splicing of pre-mRNAs, such as *Titin* and *Camk2d* (calcium/calmodulin-dependent protein kinase II delta), and therefore is required for postnatal heart development and function [78]. There is evidence that the long non-coding RNA *Malat1* is highly expressed in CMs from neonatal mice and interacts with hnRNP U in cardiac regeneration [79]. The conditional deletion of *Malat1* increases CM binucleation and reduces the window of MI-injured neonatal cardiac regeneration. The overexpression of hnRNP U in the HL-1 cardiac muscle cell line can rescue the inhibitory effect of *Malat1* deficiency on cell proliferation, raising the possibility that *Malat1* can regulate CM proliferation through hnRNP U [79]. Since the deletion of hnRNP U or *Malat1* leads to CM binucleation, there is a possibility that their interaction is required for cytokinesis, which may be important for controlling the regeneration window of injured adult hearts. However, it is not clear how *Malat1* regulates the post-transcriptional activity of hnRNP U to stimulate CM proliferation in the neonatal heart and whether the *Malat1*/hnRNP U axis modulates cardiac regeneration of the adult heart.

### 2.6. RHAU in Postnatal Heart Function and Neonatal Cardiac Regeneration

RHAU (RNA helicase associated with AU-rich element), also known as DHX36 or G4R1, possesses RNA-binding and resolvase activity to unwind the G-quadruplex (G4) structures [80]. It plays important roles during muscle development and regeneration [81]. The expression level of RHAU protein in the mouse cardiac mesoderm shows a gradual decrease after birth and becomes very low in the adult. The cardiac-specific knockout of RHAU impairs CM proliferation and causes severe heart defects, suggesting an essential role in heart development [82]. Moreover, the conditional deletion of RHAU in CMs not only leads to heart failure but also blocks neonatal cardiac regeneration after MI-induced injury, indicating that RHAU plays a role in maintaining postnatal heart function under homeostatic conditions and is required for cardiac regeneration upon heart injury [83]. As an RNA helicase that unwinds G-quadruplexes, RHAU binds to the G4 structures in the 5′-UTR (untranslated region) of *Nkx2-5*, *YAP1*, and *Hexim1* mRNAs to promote their translation [82,83]. Since Nkx2-5, YAP1, and Hexim1 are important regulators of heart development, they may function downstream of RHAU to promote CM proliferation and cardiac regeneration.

### 2.7. RBM3 and Quaking Protect Cardiomyocytes from Myocardial Apoptosis

The expression of RBM3 (RNA-binding motif 3), also known as cold-inducible RBP, is upregulated in I/R conditions; the knockdown of RBM3 in H9C2 rat CMs leads to increased apoptosis, suggesting that RBM3 should normally function to protect CM survival [84]. It is possible that RBM3 inhibits I/R injury-induced apoptosis by activating autophagy through an interaction with Raptor, a regulatory-associated protein of mTOR [84]. Although autophagy can reduce cardiac injury and preserve cardiac function under ischemia, its over-activation may produce deleterious effects for the heart under reperfusion [85]. Thus, the proper function of RBM3 may be important for autophagy activity in heart homeostasis and injury.

Quaking (QKI or QK) proteins are a family of RBPs with a STAR (signal transduction and activation of RNA) domain. QKI-5 is potentially involved in cardiovascular development and cardiac physiopathology [86]. Its expression is strongly inhibited in neonatal and H9C2 rat CMs subjected to I/R injury [87]. The overexpression of QKI-5 in I/R-injured neonatal rat CMs can prevent apoptosis by reducing the stability of FoxO1 mRNA, which encodes a proapoptotic transcription factor [87,88]. Similarly, different QKI isoforms (QKI-5, QKI-6, and QKI-7) are downregulated in the doxorubicin-treated mouse myocardium, but their overexpression can protect hearts from doxorubicin-induced cardiotoxic effects [89]. Mechanistically, QKI-5 prevents doxorubicin-induced CM apoptosis by regulating the expression of several cardiac-specific circular RNAs [89]. Thus, QKI-5 may exert protective effects on heart failure induced by chemotherapy using doxorubicin.

### 2.8. Possible Implication of RBM24 in Cardiac Repair and Hypertrophy

RBM24 (RNA-binding motif 24) is an evolutionarily conserved RBP with restricted expression and an important function during muscle development [90]. It displays dynamic cytoplasm-to-nucleus translocation during skeletal myogenesis [91], but whether this also occurs during CM differentiation remains unclear. At present, its implication in cardiac regeneration after MI or I/R injury remains unclear. However, there is evidence that it may be involved in cardiac repair and hypertrophy (Figure 5). AAV9-mediated overexpression of RBM24 in the adult mouse heart causes cardiac fibrosis, which is correlated with increased activation of cardiac fibroblasts, as well as upregulation of the ECM and immune response genes [92]. This suggests that RBM24 may function to regulate wound healing following cardiac injury. It has also been shown that RBM24 can inhibit the function of p53 by repressing the translation of its mRNA and that mice deficient in RBM24 show aberrant activation of p53-dependent apoptosis in embryonic heart tissues [93]. Nevertheless, whether RBM24 protects the survival of CMs in injured hearts is not clear. Another regeneration-related activity of RBM24 is its possible function in preventing cardiac hypertrophy through the regulation of Enh (Enigma homolog) splicing [94]. The long Enh1 isoform containing three LIM domains promotes hypertrophy, whereas the short Enh3 and Enh4 isoforms lacking these LIM domains have an inhibitory role [95]. RBM24 interacts with RBM20 to promote the alternative splicing of Enh3 and Enh4 variants, thereby repressing the hypertrophic growth of in vitro cultured rat CMs [94]. This observation indicates that RBM24 may exert a cardioprotective role. It will be interesting to understand how RBM24 functions in hypertrophy-induced heart failure.

### 2.9. PTBP1 Regulates the Reprogramming of Cardiac Fibroblasts into Induced Cardiomyocytes

The direct conversion of cardiac fibroblasts into induced CMs represents an attractive strategy for cardiac repair and regeneration, with promising potential for clinical applications [96,97,98,99]. PTBP1 (polypyrimidine tract-binding protein 1), also known as hnRNP I, is one of the most characterized repressors of cardiac-specific alternative splicing events [100]. It is highly expressed in mouse and rat embryonic hearts but is rapidly downregulated during postnatal development to become undetectable in adult hearts [101]. The overexpression of PTBP1 can induce cardiac hypertrophy and exacerbate cardiac fibrosis, either by disrupting alternative splicing or by reducing mRNA stability [102,103], while the knockdown of PTBP1 improves cardiac fibrosis in the MI-injured mouse heart, indicating a potential role in cardiac repair [102]. Importantly, PTBP1 also plays an essential role in reprogramming the cardiac cell fate. It has been shown that the expression of PTBP1 in cardiac fibroblasts prevents their conversion to induced CMs by repressing CM-specific alternative splicing patterns; conversely, the deletion of PTBP1 increases the reprogramming efficiency of induced CMs [104]. Therefore, the manipulation of PTBP1 activity in resident cardiac fibroblasts may provide an opportunity for inducing CM differentiation.

## 3. Discussion

Emerging evidence has clearly demonstrated critical roles of RBP-mediated post-transcriptional regulation in CM survival and proliferation during heart homeostasis and following cardiac injury. While many RBPs function to promote CM proliferation, others exert opposite effects or maintain cellular plasticity (Table 1). Obviously, RBPs are immediate effectors of extrinsic or intrinsic injury stimulus and subsequently function as key regulators of cardiac regenerative capacity or hypertrophic growth. As effectors, it will be intriguing to understand how cardiac injury-stimulated signals regulate the expression of various RBPs, at transcriptional, post-transcriptional, translational, and post-translational levels, because it seems that all of these mechanisms should be involved in coordinating the expression or activity of RBPs under homeostatic and pathological conditions. Several signaling pathways, such as Wnt/ß-catenin, are important in heart regeneration [105]. How they regulate RBPs or control the outcome of RBP-mediated cardiac regeneration merits future investigation. In addition, cardiac stem cells have the potential to regenerate injured CMs [106]. It is therefore of interest to understand the interplay between stem cell factors and RBPs in CM proliferation and differentiation. As regulators, several aspects have appealed increasing interest. First, RBP-mediated translational activation/repression appears to be an important mechanism triggering cardiac growth and regeneration because the adult heart shows surprisingly low rates of protein synthesis. Indeed, key components of the cytoplasmic polyadenylation complex, such as PABPC1 and CPEB4, are involved in cardiac hypertrophy [18,68]. Interestingly, RBM24 also has the potential to repress the hypertrophic growth of CMs [94], and it can function in regulating mRNA translation during cellular differentiation [107]. Thus, the role of cytoplasmic polyadenylation in reprogramming the translational landscape after cardiac injury merits further investigations. Second, the interaction of RBPs with non-coding RNAs adds another layer of regulation for repairing the injured heart. Indeed, accumulating evidence suggests that non-coding RNAs also function as important post-transcriptional regulators of CM proliferation and present strong potential as therapeutic targets [108,109,110]. It is of note that RBPs and non-coding RNAs mutually regulate the biogenesis and function of each other, thereby ensuring CM survival and proliferation following cardiac injury. Third, an important question in studying cardiac regeneration is to provide evolutionary insights into the differential regenerative capacity among vertebrate species. It is well established that adult mammalian hearts show no or very limited regeneration after injury. This is at least partly to the lack of sufficient mononuclear diploid CMs, which represent a relatively small subpopulation in the adult heart and are responsible for the differential regenerative capacity between zebrafish and mammals [32,33,111,112,113]. Consistently, the induction of polyploidization in zebrafish and mouse neonatal hearts prevents cardiac regeneration, whereas a reduction in polyploidization delays cell cycle exit and promotes CM proliferation, thereby retaining the regenerative potential of adult hearts [114,115,116,117]. Interestingly, several RBPs are implicated in CM binucleation either during cardiac regeneration, such as LIN28A and hnRNP U [26,79], or during heart development, such as RBPMS, which is not discussed here [118]. Thus, it will of interest to further understand the post-transcriptional regulatory mechanism by which they dictate CM ploidy during heart development in different vertebrates. Last but not least, the dynamic expression of RBPs in different cardiac cell types is also essential for the regulation of cellular plasticity. In this regard, the activity of MBNL1 is clearly involved in the transformation of fibroblasts into myofibroblasts and in the proliferative capacity of CMs [74,75,76,77]. Furthermore, manipulating the repressive activity of PTBP1 on CM-specific alternative splicing can increase the reprogramming efficiency of cardiac fibroblasts into induced CMs [104]. Together, many lines of evidence have unequivocally identified RBPs as important effectors and regulators of cardiac development and regeneration. However, given the key roles of RBP-mediated post-transcriptional regulation in lineage differentiation and tissue homeostasis, the implications, as well as biochemical and functional interactions, of many other cardiac-specific RBPs in cardiac repair and regeneration merit further investigations.

## 4. Conclusions

There is a growing interest in understanding the post-transcriptional regulation of cardiovascular disease. It is clear that RBPs are important effectors that relay extrinsic or intrinsic signals to reprogram cellular activity and cell fates after cardiac injury. The interplay between various RBPs contributes to maintaining cardiac homeostasis and modulating cardiac regenerative capacity. Several RBPs, such as LIN28A, MBNL1, PTBP1, and m6A readers, are important regulators of cardiac growth and plasticity, representing promising therapeutic targets for modulating CM proliferation and differentiation during regeneration. Therefore, the identification of cardiac-specific RBPomes, as well as dynamic changes in the binding activity of RBPs between homeostatic and injured conditions, will be important for uncovering regeneration-associated post-transcriptional events. A comprehensive analysis of the interplay between different RBPs in CM proliferation should provide further mechanistic insights into adult mammalian cardiac regeneration.

## Figures and Tables

**Figure 1 ijms-24-12004-f001:**
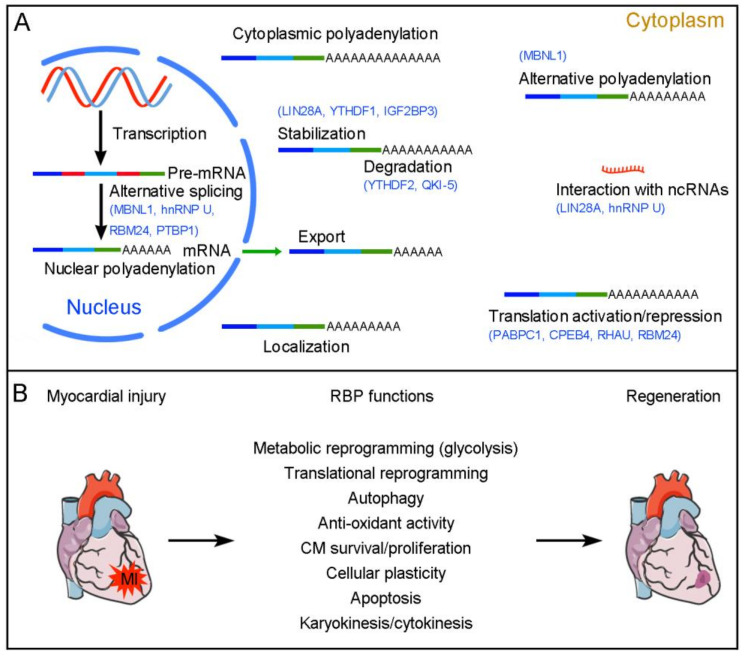
RBP-mediated post-transcriptional regulation of cardiac regeneration. (**A**) RBPs regulate pre-mRNA alternative splicing in the nucleus and control mRNA polyadenylation, export, localization, stability/degradation, and translation in the cytoplasm. Several RBPs with known post-transcriptional regulatory functions in cardiac regeneration are indicated in blue. (**B**) Functions of RBPs in cardiac regeneration. RBPs are either activated or repressed following cardiac injury. They coordinate regeneration-associated post-transcriptional events to modulate the formation of new CMs in injured neonatal and adult hearts.

**Figure 2 ijms-24-12004-f002:**
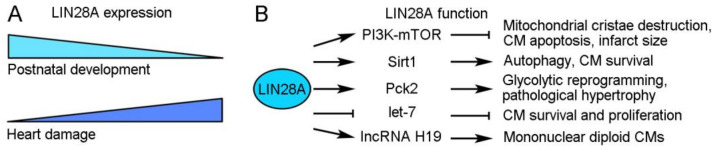
LIN28A expression and function during cardiac regeneration. (**A**) LIN28A expression during postnatal development and heart damage. (**B**) Regulatory roles of LIN28A in cardiac regeneration.

**Figure 3 ijms-24-12004-f003:**
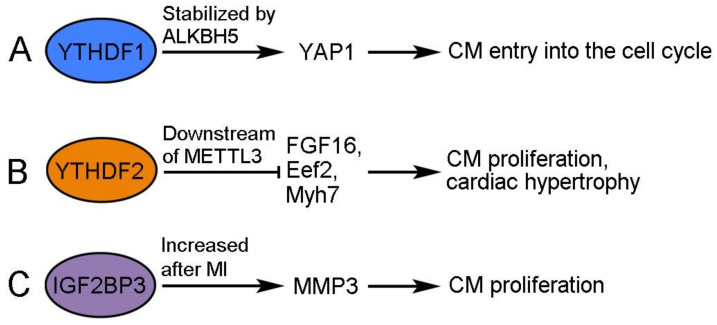
Regulation and function of m6A readers in CM proliferation. (**A**) YTHDF1 promotes the re-entry of CMs into the cell cycle. (**B**) YTHDF2 functions downstream of METTL3 to prevent CM proliferation. (**C**) IGF2BP3 promotes CM proliferation by regulating MMP3 expression.

**Figure 4 ijms-24-12004-f004:**
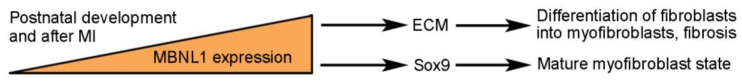
MBNL1 regulates cellular plasticity in cardiac regeneration.

**Figure 5 ijms-24-12004-f005:**
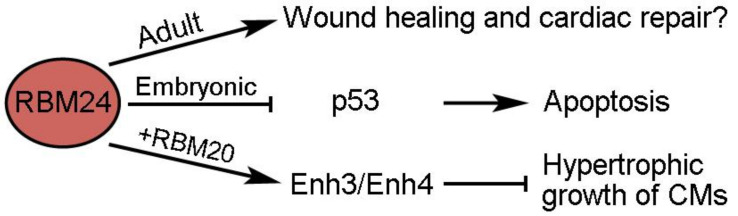
Potential functions of RBM24 in cardiac repair and homeostasis.

**Table 1 ijms-24-12004-t001:** Roles of RBPs in cardiac regeneration.

RBPs	Expression	Functions	Targets	References
LIN28A	Decreases in postnatal heart; rapidly induced after heart injury	Metabolic changes, autophagy, anti-oxidant activity, mononucleation, CM proliferation	Insulin–PI3K–mTOR pathway, *Pck2* and *Sirt1* mRNAs, *H19* lncRNA, *let-7* miRNA	[25,26,27,28,29,30,34]
YTHDF1	Decreases in postnatal heart; increases in injured neonatal heart	Triggers CMs to re-enter cell cycle and promotes cardiac regeneration	*YAP1* mRNA	[47]
YTHDF2	Increases in heart failure	Mediates the effects of METTL3 to prevent CM proliferation	*FGF16*, *Myh7*, and *Eef2* mRNAs	[50,51,52,53]
IGF2BP3	Decreases in postnatal heart; increases in injured neonatal heart	Promotes CM proliferation and cardiac regeneration	*MMP3* mRNA	[55,56]
PABPC1	Protein absent in the adult heart but rapidly accumulates after injury	Promotes CM growth during cardiac hypertrophy	Interacts with eIF4G to induce global protein synthesis	[68]
CPEB4	Unchanged between normal and injured hearts	Prevents pathological cardiac hypertrophy	*Zeb1* mRNA	[18]
MBNL1	Increases in postnatal and injured hearts	Prevents CM proliferation; maintains a mature myofibroblast state	*Sox9* mRNA	[75,76,77]
hnRNP U	Decreases in postnatal and adult hearts	Promotes CM proliferation	Interacts with *Malat1* lncRNA	[78,79]
RHAU	Decreases in postnatal hearts	Promotes CM proliferation	*Nkx2-5*, *YAP1*, and *Hexim1* mRNAs	[82,83]
RBM3	Increases after heart injury	Promotes CM survival	Interacts with Raptor	[84]
QKI-5	Inhibited after heart injury	Protects CMs from apoptosis	*FoxO1* mRNA, cardiac-specific circular RNAs	[87,88,89]
RBM24	Highly expressed in embryonic and adult hearts	Promotes cardiac fibrosis and represses CM hypertrophic growth	ECM and immune response genes, *p53* mRNA, *Enh* splicing	[92,93,94]
PTBP1	Decreases in postnatal heart; undetectable in adult heart	Induces cardiac hypertrophy and fibrosis; regulates reprogramming of cardiac fibroblasts into induced CMs	CM-specific alternative splicing	[101,102,103,104]

## Data Availability

No new data were created in this work.

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
