# Peer review of "RNA-Binding Proteins as Critical Post-Transcriptional Regulators of Cardiac Regeneration"

_ijms, 2023, doi:10.3390/ijms241512004_

Round 1

Reviewer 1 Report

This review is interesting and useful to specialists, it contains up-to-date information.

The review is well structured, the main information is summarized in the final table.

However, there is one drawback: the text is very difficult to read due to too much information.

To improve understanding, I suggest adding small schemes to chapters  2.1, 2.2.1, 2.2.2, 2.2.3, 2.3, 2.4, 2.6, 2.8. I propose to summarize the role of protein in the cardiac regeneration (including protein's partners and effects of interactions) on these schemes.

Author Response

This review is interesting and useful to specialists, it contains up-to-date information.

The review is well structured, the main information is summarized in the final table.

However, there is one drawback: the text is very difficult to read due to too much information.

To improve understanding, I suggest adding small schemes to chapters 2.1, 2.2.1, 2.2.2, 2.2.3, 2.3, 2.4, 2.6, 2.8. I propose to summarize the role of protein in the cardiac regeneration (including protein's partners and effects of interactions) on these schemes.

Response:

Thank you for the positive assessment of the manuscript. I followed your helpful suggestion by adding small schemas to the corresponding sections. No schema is added to section 2.4 because this section is short and describes simply a requirement of RHAU for CM proliferation likely through regulation of Nkx2-5, YAP1, and Hexim1.

Reviewer 2 Report

Please see attached PDF.

Author Response

General Comments:

The author is to be commended for tackling a complex subject and one as important as cardiac regeneration. Studies on cardiac stem cells have produced very little of therapeutic value over the last 15 years so other methods of initiating native stem cell regeneration may be of use in the fight against ischemic heart diseases. Some minor issues, specified below, are detailed and need to be fixed before publication. The language used in the manuscript is suitable for the topic and only minor, insignificant issues are present. Minor revisions are recommended.

Response:

Thank you for the enthusiasm and positive evaluation of this work.

Specific Comments

Introduction:

Lines 31-49: The introduction focuses entirely on cardiovascular disease but does not detail the types of myocardial issues and their putative causes. Ischemia from a birth defect that requires surgical repair can cause a different set of lifelong sequelae than ischemia from COPD or other lifestyle-mediated myocardial diseases. It is recommended to make a small table with the type of myocardial damage and potential RBPs/activators of RBPs involved. This will set up the rest of the paper very nicely. Use “unknown” if an RBP activator or other item is not currently reported; such information will also highlight the need for further researches into RBP regulation.

Response:

Thanks for the insightful suggestion. Since most studies have been focused on I/R-induced and apical resection injuries in animal models, it is not easy to present an informative table here. However, this issue was addressed in the revised manuscript by presenting different types of myocardial injury with their causes by citing two relevant references. This section also mentioned the analysis of RBPs in ischemia-dependent and -independent myocardial injury using different animal models.

Figure 1 is welcome and explains RBP concept well.

Response:

Thanks

What is the role of quality control in the RBP process? There needs to be an initial and general explanation as to the temporal position RBPs take in the post-translational machinery. Do RBPs act before or after folding? Do they assist in folding? Do they participate in the Ub/Prot or autophagy systems? Please make this clear in a few sentences so that the exact utility of RBP regulation as a therapeutic process can be generally localized before the specific RBPs are explained.

Response:

Thanks for the helpful comment. In the revised manuscript, the role of RBPs in protein quality control is explained by citing a reference.

Manuscript Body:

Of importance: Why were these specific RBPs chosen? Are they myocardium-specific or are they present in other types of cells?

Response:

This review discusses the known functions of RBPs in cardiac repair and regeneration. This is highlighted in the revised manuscript. Some of these RBPs are cardiac-specific while others are also expressed in other cell types.

Line 133-138: The effect of LIN28A in facilitating hypertrophy in adults vs. regeneration in natal stages needs explanation on the differences. Since the gene/protein itself is not changed, is the sole action of let-7 miRNA the key factor in determining the role of regeneration vs. hypertrophy? Is the oxygen milieu a key determiner of the expression of let-7? And do the Nrf2-mediated antioxidant response or second-messenger reactive oxygen species systems play any role?

Response:

As a pluripotent cell factor, the function of LIN28A in cardiac growth is complex. Mechanistic insights underlying the function of LIN28A in regeneration and hypertrophy await further investigation. This is discussed in the revised manuscript. At present, there is no evidence on the direct regulation of let-7 expression by oxygen and the Nrf2-mediated antioxidant response or second-messenger reactive oxygen species.

Line 261-316: This is out of place in a detailed description of individual RBPs. Suggest moving it to the beginning of section 2 as an example process mediated by RBPs to highlight the mechanistic pathways that RBPs are involved in.

Response:

In response to this suggestion, the title of this section is changed to “Translational reprogramming mediated by PABPC and CPEB in cardiac hypertrophic growth”, such that it fits with a detailed description of individual RBPs.

2.4, 2.5, 2.6 titles are all in italics, even though, previously, the individual descriptions of RBPs are not. Please be consistent.

Response:

According the journal template, the heading is in italic but the subheading is not.

Table1 is quite welcome and detailed; however, are they listed in some kind of special order? If not, it is suggested to group them based on their roles/positions in mechanistic pathways within the table for ease of reference.

Response:

RBPs in table 1 are listed in the order as described in the text. Since many RBPs display multiple as well as redundant and distinct functions, it is not easy to group them in a table based on their roles.

Discussion:

The Discussion must mention the role of cardiac stem cells and if Yamanaka (e.g., KLF4), neural crest stem cell factors (e.g., SOX-family factors), or other factors involved in these stem cells also play a role in either regulating RBPs or controlling the outcome of RBP-regulated cardiac myocytes in the differentiation process. The role of cmyc and WNT should also be detailed here…because all developmental biologists will be looking for it.

Response:

Thanks for these recommendations. These aspects were discussed in the revised manuscript.

How does exercise regulate RBP expression and activity, especially with respect to cardiac remodeling after ischemia? This needs to be explained, since exercise is a potent regulator of the miRNA that regulate remodeling.

Response:

An example of increased expression and activity of PABPC1 by endurance exercise training was provided in section 2.3.

Conclusions:

These conclusions are too general. Specific recommendations on RBPs are required to focus future research onto pathways that would be useful in regenerating the myocardium. Which RBPs show the most promise, based on current evidence, of playing a key role in therapeutic development?

Response:

Thanks for this suggestion. Several RBPs that function as important regulators of cardiac growth and plasticity with promise in therapeutic development were proposed in this section.
